# An Efficient Parallel Algorithm for Detecting Packet Filter Conflicts

Chun-Liang Lee [1] , Guan-Yu Lin [2,*] and Yaw-Chung Chen [2]

1   Department of Computer Science and Information Engineering, School of Electrical and Computer
    Engineering, College of Engineering, Chang Gung University, Tao-Yuan 33302, Taiwan; cllee@mail.cgu.edu.tw
2   Department of Computer Science, National Yang Ming Chiao Tung University, Hsinchu 30010, Taiwan;
    ycchen@cs.nctu.edu.tw
*   Correspondence: guanyu@cs.nctu.edu.tw

**Abstract:** Advanced network services, such as firewalls, policy-based routing, and virtual private networks, must rely on routers to classify packets into different flows based on packet headers and predefined filter tables. When multiple filters are overlapped, conflicts may occur, leading to ambiguity in the packet classification. Conflict detection ensures the correctness of packet classification and has received considerable attention in recent years. However, most conflict-detection algorithms are implemented on a conventional central processing unit (CPU). Compared with a CPU, a graphics processing unit (GPU) exhibits higher computing power with parallel computing, hence accelerates the execution speed of conflict detection. In this study, we employed a GPU to develop two efficient algorithms for parallel conflict detection: the general parallel conflict-detection algorithm (the GPCDA) and the enhanced parallel conflict-detection algorithm (the EPCDA). In the GPCDA, we demonstrate how to perform conflict detection through parallel execution on GPU cores. While in the EPCDA, we analyze the critical procedure in conflict detection as to reduce the number of matches required for each filter. In addition, the EPCDA adopts a workload balance method to enable load balancing of GPU execution threads, thereby significantly improving performance. The simulation results show that with the 100 K filter database, the GPCDA and the EPCDA execute conflict detection 2.8 to 13.9 and 9.4 to 33.7 times faster, respectively, than the CPU-based algorithm.

**Keywords:** conflict detection; GPU; packet classification

## 1. Introduction

Packet classification techniques play a significant role in providing advanced network services, such as packet filtering, quality of services (QoS), security monitoring, and virtual private networks [1]. The packet classifier runs on the Internet router to classify the received packets into different flows based on predefined rules called packet filters. Different network services may use different header fields to classify the packets. For example, in IPv4, five header fields are typically used in packet filters; these include the source/destination Internet protocol (IP) address, the source/destination port, and the protocol type.

A filter, $F$, with $d$ fields is called a $d$-dimensional filter, expressed as $F = (f[1], f[2], ..., f[d])$, in which the content in the $i$th field $f[i]$ could be a variable-length prefix, an exact value, a range, or a wildcard, indicating that all values are valid for that field. For packet $P$ and filter $F$, if all selected packet headers correspond to the values in their associated fields in $F$, we say that packet $P$ matches filter $F$. For example, a two-dimensional (2D) filter $F = (140.113. *. *, *)$, source IP address (SA) with prefix 140.113.xxx.xxx, and any destination IP address (DA) will match $F$. Therefore, packet $p_1 = (140.113.1.1, 8.8.8.8)$ matches $F$, but $p_2 = (140.114.1.1, 8.8.8.8)$ does not. Each filter exhibits an associated action specifying how to treat those packets that match the filter. When a packet matches multiple filters with different actions, filter conflict occurs, resulting in ambiguity in packet classification. For example, Table 1 presents a 2D filter database for firewall applications. Assuming that the

IP address length is four bits, incoming packet $p_1$ = (0001, 1000) will match filter $a$ and be accepted. Another incoming packet, $p_2$ = (0001, 0000), matches filters $a$ and $b$. Because $a$ and $b$ exhibit different actions, they cannot decide whether the packet should be accepted or rejected, resulting in a conflict. The conflicting actions of $a$ and $b$ may raise security vulnerabilities, QoS failures, or routing errors [2,3], depending on the application used. Three possible solutions can be applied to solve the conflict problem [3].

**Table 1.** Example of 2D filter database.

| Filter | Source IP Address | Destination IP Address | Action |
|--------|------------------|------------------------|--------|
| $a$ | 00* | * | Accept |
| $b$ | * | 00* | Reject |
| $c$ | 11* | * | Accept |
| $d$ | * | 11* | Reject |

1. Select the first matching filter in the filter database.
2. Assign each filter a priority and select the matching filter with the highest priority.
3. Assign each field a priority and select the matching filter with the most specific matching field with the highest priority.

However, none of the above methods can fully solve the conflict problem. Figure 1 depicts a 2D representation based on the contents of Table 1, and the overlapped areas indicate conflicts between filters. Let $a \rightarrow b$ indicate that when a packet matches filters $a$ and $b$ simultaneously, the action associated with filter $a$ is selected. In other words, filter $a$ has a higher priority than filter $b$. If we set $b \rightarrow c$, $c \rightarrow d$, and $d \rightarrow a$, we observe that priority setting may lead to $a \rightarrow a$, which is a contradiction. There is no way to find a priority sequence with no conflicts. Therefore, resolve filters [3] have been developed to solve the filter-conflict problem. The idea is to add a new filter to the overlapped area of two conflicting filters and set a higher priority to resolve the conflict. In Figure 1, resolve filter $e$ is generated for the overlapped area of filters $a$ and $b$. Similarly, the other three overlapped areas require associated resolve filters. For every resolve filter generated, it is necessary to ensure that it does not conflict with the other filters. In addition, some applications must update packet filters frequently [4]. For example, within one millisecond, several access controls or QoS filters may be updated, or several thousand filters may be changed, due to the dynamic creation or the deletion of the virtual routers [5]. Therefore, conflict detection must be executed for every newly added or updated filter to prevent conflicts and ensure the correctness of the packet classification. Consequently, the efficiency of conflict detection influences network performance. It has been demonstrated that determining the minimum number of resolve filters in a filter database is an NP-hard problem [3,6].

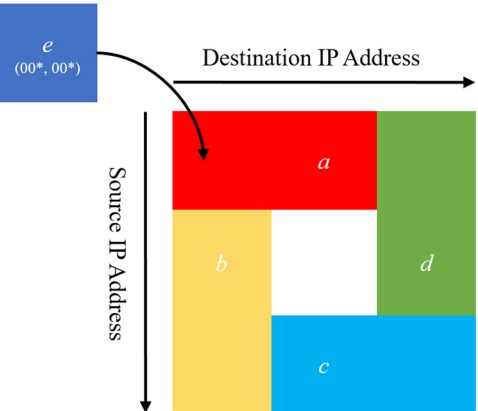

**Figure 1.** Rectangular presentation of filters.

Diversified network services will appear continuously, due to the development trend of the Internet, and the transmission speed and data volume will continue to increase. Regarding the filter database in routers, the number of filters will increase significantly. In addition, due to the deployment of IPv6 and the development of software-defined networks [7], filters should deal with more content, and the number of fields to match in conflict detection is much greater than the number of fields in traditional network architecture. These new types of network protocols or services may increase conflict-detection complexity. Currently, most conflict-detection algorithms can only handle 2D conflict detection, and for practical operation, additional dimensions other than the source/destination IP address must be included. Although some 2D conflict-detection algorithms can be extended to handle more dimensions for conflict detection, they require a large memory space. Therefore, designing an efficient conflict-detection algorithm is still a challenge.

In packet classification applications, the classification process for each packet is independent. Due to the development trend of CPUs, researchers have investigated packet classification on multicore CPUs [8–12]. However, from the viewpoint of parallelism, the number of cores is not sufficiently high, and the scale of performance improvement is limited. Compared with a CPU, a general-purpose GPU has a large number of cores and can offer a superior parallel computing capability. Because the reduced dependence of data and control makes them more appropriate for the parallelism on multi-core and many-core systems. Many researchers have attempted to use GPUs to solve computing intensive problems in related domains [13–25]. Similarly, each filter is independent while executing conflict detection, and the field content matching does not require complex computations. Thus, conflict detection can be executed in parallel to improve performance significantly.

In this study, we propose two parallel algorithms that can solve five-dimensional (5D) filter conflict problems. First, we formally defined the conflict-detection problem. To develop efficient conflict-detection algorithms, we divided a 5D filter into two parts based on the field format. We then derived the conditions under which both filters must hold if they conflict. Based on these conditions, we developed a filtering mechanism that reduced the number of comparisons for each filter during conflict detection. With the proposed filtering mechanism, each filter experienced different comparisons, resulting in inefficient parallel processing in the GPU because of workload imbalance among threads. Therefore, we propose a scheme for workload balancing to further improve the parallel computing performance.

The remainder of this paper is organized as follows. Section 2 reviews existing conflict-detection algorithms and briefly introduces the GPU architecture. In Section 3, we propose a simple 5D conflict-detection algorithm. In Section 4, we present our proposed general parallel conflict-detection algorithm (the GPCDA) and enhanced parallel conflict-detection algorithm (the EPCDA). Section 5 describes the experimental setup, results, and analyses. Finally, Section 6 concludes this study.

## 2. Related Work

### 2.1. Existing Conflict-Detection Algorithms

A straightforward approach to detecting all filter conflicts is to check every pair of filters in the filter database. This approach does not utilize extra storage to cache additional information and makes it easy to detect conflicts. However, it takes $O(n^2)$ time to detect all the conflicts, where $n$ denotes the number of filters. Hari et al. [3] defined the source IP address and destination IP address fields as 2D prefix fields. They introduced the filter conflict concept and proposed an algorithm for detecting all conflicts in 2D prefix filters. We assume that the filter $f$ has two prefix fields. Let $f[1]$ and $f[2]$ denote the first prefix field and the second prefix field, respectively. Filters $f$ and $g$ conflict when any of the following two conditions hold.

1. $f[1]$ is a prefix of $g[1]$, and $g[2]$ is a prefix of $f[2]$.
2. $g[1]$ is a prefix of $f[1]$, and $f[2]$ is a prefix of $g[2]$.

Hari et al. [3] developed the *FastDetect* algorithm to perform 2D prefix filter conflict detection using two grid-of-tries and switch pointers [26]. All conflicts can be detected within $O(nW + S)$ time, where $n$ is the number of filters, $W$ is the length of the prefix, and $S$ is the number of conflict pairs generated by the filters. In addition, the space complexity is $O(nW)$. When the *FastDetect* algorithm is extended to 5D conflict detection, the source/destination port and protocol fields are included for comparison. The preset content of non-IP address fields can only be an exact value or a wildcard. This does not conform to practical applications.

Baboescu and Varghese [4] proposed the scalable bit vector (SBV) conflict-detection algorithm based on the bit vector (BV) scheme [27] and the aggregated bit vector (ABV) scheme [28]. They used a compressed binary trie in each field. Each node in tries needs to store an $n$-bit vector. In a static mode, their proposed algorithms can detect all conflicts with time complexity $O(knW)$, where $k$ is the number of fields. In addition, the space complexity is $O(kN^2)$. When dealing with 5D conflict detection, the data structure of the three binary tries must be added to process the information of the three non-IP address fields. When the port field is specified by a range, range-to-prefix conversion is required, leading to significant filter duplications that consume large memory space. Lai and Wang [29] established several algorithms that modified the original BV scheme to prevent the defect of massive memory duplication caused by conversion from range to prefix and developed a method for comparing range fields. This allows the algorithm to support 5D conflict detection. However, it still incurs a high cost in memory space. Lee et al. [30] proposed an algorithm to improve the SBV algorithm by reducing the amount of bits required to be read when bit vectors are accessed. They divided all filters into several groups based on prefix fields, and then constructed a conflict matrix to indicate whether two groups were in conflict or not. The experimental results showed that their algorithm reduced the average detection time per filter by up to 77.9%, compared with the SBV algorithm. However, this algorithm focuses on 2D filter databases, and cannot be easily extended to handle 5D filter databases. Kuo et al. [31] proposed a compact bit vector (CBV) conflict-detection algorithm to improve the SBV algorithm. First, they proposed a redundancy reduction scheme that explored and exploited the covering and potential conflict relations between filters to significantly reduce the number of filters that must be involved in the construction of matching tries. Second, the CBV algorithm further merged the redundant match nodes in each matching tries by adopting an upward merging approach. Finally, the highly compact matching tries were built to represent the relationships between filters. As with the data structures used in [4,29,30], using the trie-based data structure incurs high memory usage.

Lu and Sahni [6] discovered that when 2D filters are translated to geometric area representation, if two filters are conflicted, their represented area will cause overlap in the plane, and the line segment belonging to the two areas may have a perfect crossing. Therefore, they designed a magnifying mechanism to ensure that each filter conflict exhibited a perfect crossing. They used a plane sweep method [32] to determine all perfect crossings in the plane, and through the position, filter conflicts could be detected. The algorithm had a time complexity $O(n\log n + S)$ and space complexity $O(n)$. The experimental results showed that for five filter sizes with various prefix lengths of filter, their proposed algorithm was 4 to 17 times faster than [3], with 4 to 8 times less memory requirement, and was 4 to 27 times faster than [4], with 6 to 205 times less memory requirement. Lee et al. [33] studied the tuple-space search algorithm [34], observed the relationship between tuples, derived the relationship for searching among tuples by generating markers, and proposed a tuple-based conflict-detection algorithm (TCDA). The execution of conflict detection in each filter could be accelerated by adding a marker pointer and a filter pointer. Finally, the TCDA can detect all conflicts with time complexity $O(nW + S)$ and space complexity $O(nW)$. Compared with [6], the experimental results showed that the proposed algorithm can reduce the detection time by 19.6% to 97.7%. More importantly, it reduced the storage requirements of most filter databases. The performance improvement over time is particularly significant for filter databases with many conflicts. Neither [6] nor [33] proposed a method for 5D

conflict detection. Yi et al. [35] used a formal method to analyze the meaning of IPv6 firewall filters and developed a method to find the filter conflicts. However, that research did not illustrate how to support 5D conflict detection.

### 2.2. CUDA Architecture and Programming Model

In 2007, NVIDIA introduced the Compute Unified Device Architecture (CUDA), which provides a platform for developing parallel computations on GPUs. The CUDA defines parallel thread execution and instruction set architecture so that GPU cores in the CUDA can process a specific segment of a parallel program simultaneously. The CUDA provides libraries and toolkits for developers, and version 11.6.2 [36] is currently the most updated. Following the evolution of hardware from version 1.0 to version 8.7, the GPU architecture provides increased GPU cores and computing power to improve GPU computing efficiency. The streaming processor (SP) is the basic processing unit. Many SPs in a GPU can perform computations simultaneously, and several SPs attached to other units, such as memory, computing units, and control units, can form a streaming multiprocessor (SM). An SM is the basic hardware unit for executing threads in the CUDA. CUDA-capable GPUs have a memory hierarchy, as shown in Figure 2. The GPU architecture exposes the following memory units.

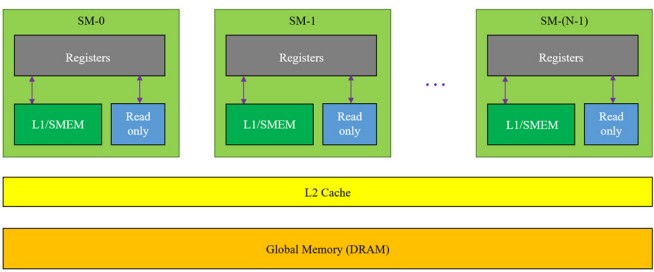

**Figure 2.** Memory hierarchy in GPUs.

1. Registers: These are private to each thread, suggesting that registers assigned to one thread are invisible to other threads.
2. L1/Shared memory (SMEM): Every SM has a fast, on-chip scratchpad memory that can be used as an L1 cache and SMEM. All threads in a CUDA block can share shared memory, and all CUDA blocks running on a given SM can share the physical memory resources provided by the SM.
3. Read-only memory: Each SM has an instruction cache, constant memory, texture memory, and a read-only cache.
4. L2 cache: The L2 cache is shared across all SMs; therefore, every thread in every CUDA block can access this memory.
5. Global memory: As with the L2 cache, all threads can read and write this memory, but it is slower to access than other memories. This memory can be used to exchange data between the GPU and the CPU.

A CUDA program contains two parts: the host function and device function, which are executed on the CPU and the GPU, respectively. Initially, the host function copies the data required by the device function to the device memory of the GPU through PCI-express and executes the device function on the GPU. The host then retrieves the results from the device memory when GPU computation is performed. The minimum execution unit in the device is a thread; several threads form a block, and several blocks form a grid. Before executing a program, program developers can set the number of threads and blocks based on the purpose of the computation. The maximum number of threads allowed within a block varies, depending on the hardware version. Up to 1024 threads are allowed in the current version. While the device is in execution, it assigns blocks to the SMs for computation block by block, and threads in the block are executed in groups based on the warps. Figure 3 illustrates an actual execution in the CUDA. Each warp consists of 32 threads, and the

threads in each warp perform computations simultaneously. When an SM provides more warps, more threads can be executed simultaneously. The maximum number of warps supported by an SM varies, depending on the hardware specification.

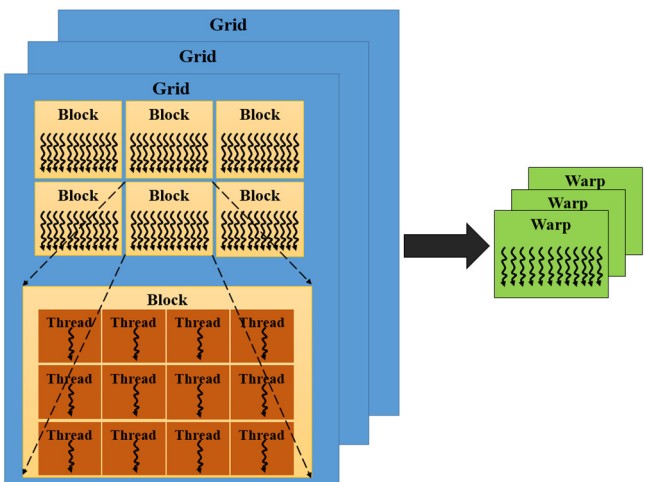

**Figure 3.** Programming model of the CUDA.

Each filter is mutually independent when executing conflict detection; therefore, we can assign the task of conflict detection to the device for execution. The task is then distributed to the threads for computation to improve performance through parallel computing. However, this distribution may reduce parallel computing performance if the workload distribution is imbalanced. For example, when more workload is assigned to fewer threads, the remaining threads accomplish the work earlier because of fewer assigned tasks, and these threads become idle. Therefore, evenly distributing the workload to threads is a critical issue in parallel computing.

## 3. Definition and Method of 5D Conflict Detection

In this study, we define 5D conflict detection as the matching of five common fields in an IP header. The five fields are SA, DA, source/destination port (SP/DP), and protocol. Table 2 presents an example of a 5D filter database. In this section, we define how the two 5D filters encounter the conflict condition, then we explain our proposed simple and fast 5D filter detection algorithm.

**Table 2.** Example of a 5D filter database.

| Filter | SA | DA | SP | DP | Protocol |
|--------|------|------|------|--------|----------|
| $F_0$ | 101* | 01* | * | 0–1023 | TCP |
| $F_1$ | 10* | 010* | * | 80 | * |
| $F_2$ | 10* | 0* | * | * | UDP |
| $F_3$ | 10* | 0* | * | 0–1023 | * |

### 3.1. Definition of 5D Filter Conflict

Conflict detection is relatively easier in 2D because it uses only two fields. If the conflict condition defined in [3] is extended to 5D conflict detection, it should compare five fields. When two filters, $f = (f[1], f[2], \ldots, f[5])$ and $g = (g[1], g[2], \ldots, g[5])$, satisfy the following two conditions, we can determine that filter $f$ conflicts with filter $g$.

1. $\forall_{1 \leq k \leq 5}(f[k] \cap g[k]) \neq \varnothing$.
2. $\exists_{1 \leq x \leq 5}\exists_{1 \leq y \leq 5}((x \neq y) \wedge (f[x] \subset g[x]) \wedge (g[y] \subset f[y]))$.

Condition 1 indicates that when both filters $f$ and $g$ find any field $k$ such that the set of matched results is empty, no conflict exists between filters $f$ and $g$. For example, when

matching the protocol field of filters $F_0$ and $F_2$ (Table 2), the result is empty because no packet uses the transmission control protocol and the user datagram protocol simultaneously. In other words, when all fields $k$, such as the matching result of $f[k]$ and $g[k]$, are not empty, $f$ and $g$ may have conflicts. If we can find fields $x$ and $y$ such that the results of matching filters $f$ and $g$ satisfy condition 2, we can conclude that $f$ and $g$ have conflicts. For example, all field matching for filters $F_1$ and $F_2$ are nonempty (Table 2), and we can always find DA and protocol fields such that $F_1[DA] \subset F_2[DA]$ and $F_2[Protocol] \subset F_1[Protocol]$. Therefore, $F_1$ and $F_2$ are conflicted.

*3.2. Conflict Detection for 5D Filters*

If two matching filters satisfy condition 1, we select any two fields to fulfill condition 2; we have 10 combinations of fields. Furthermore, because of the different formats of stored data for different fields, the conflict-detection process becomes complex and tedious if we match every combination individually. In addition, it is challenging to design an efficient data structure and algorithms to perform 5D conflict detection. Furthermore, the performance of GPU parallel computing is limited.

To reduce the complexity of 5D conflict detection, we designed a simple detection algorithm that separated the 5D conflict-detection process into two parts, the matching of prefix fields (SA/DA, two fields) and non-prefix fields (SP/DP/Protocol, three fields). When matching two filters to verify whether they are conflicted, it first checks the prefix field matching result and matches the non-prefix field contents to determine whether a conflict will occur. By excluding the empty set, the matched result of prefix fields $i$ and $j$ contains three cases. For each case, we describe the combination of the non-prefix field content of filters $f$ and $g$ for conflict to occur.

1.  $((f[i] \subset g[i]) \wedge (g[j] \subset f[j]))$: This case indicates that the prefix fields of filters $f$ and $g$ conflict, as defined by [3]. Therefore, when the matched results of the non-prefix fields of $f$ and $g$ are all nonempty sets, $f$ conflicts with $g$. The filter pair $(F_0, F_1)$ belongs to this case because the non-prefix field matched results of $F_0$ and $F_1$ are all nonempty sets (Table 2).

2.  $((f[i] \subseteq g[i]) \wedge (f[j] \subset g[j]))$: This case indicates that the prefix fields of $f$ have at least one field content contained in $g$. Thus, if the matched result of non-prefix fields of $f$ and $g$ are all nonempty sets, and there is a field $s$ such that $(g[s] \subset f[s])$, $f$ conflicts with $g$. Both filter pairs, $(F_0, F_3)$ and $(F_1, F_2)$, belong to this case (Table 2); therefore, conflicts occurred in both cases. However, in the matching of non-prefix fields in $(F_0, F_3)$, it did not find any field $s$ such that $(F_3[s] \subset F_0[s])$. Thus, $(F_0, F_3)$ did not have conflict.

3.  $((f[i] = g[i]) \wedge (f[j] = g[j]))$: This case suggests that the contents in the prefix fields of $f$ and $g$ are equal. Therefore, when the matched results in the non-prefix fields of $f$ and $g$ are all nonempty, and we find any two fields $s$ and $t$ such that $((g[s] \subset f[s]) \wedge (f[t] \subset g[t]))$, then $f$ and $g$ have conflict. The filter pair, $(F_2, F_3)$, is categorized into this case because it can find the DP and protocol fields that satisfy condition 2, as defined in Section 3.1.

Based on the above analysis, all filter pairs with conflicts should satisfy one of the above cases. In other words, we can focus on the above analysis to design a simple detection algorithm that can replace the original complicated matching process. The execution steps of the detection algorithm are as follows.

1.  It determines whether the matching result of the prefix fields of the filters satisfies any of the cases mentioned above. If it does, execute step 2; otherwise, terminate the process.

2.  It matches the non-prefix field content based on the cases satisfied and then determines whether the matched result is the case for conflict to occur.

## 4. Conflict-Detection Algorithms Using GPU

In this section, we propose two new 5D detection algorithms implemented on GPUs. First, we introduce the implementation of the detection algorithm on a GPU and describe its operational process. The number of matches required by each filter varies to prevent

a duplicate detection process because the workload for each filter is different. The main design concept of the EPCDA is the proper assignment of filters to threads for balancing the workload of each thread and improving the parallel computing performance.

### 4.1. General Parallel Conflict-Detection Algorithm

During conflict detection, the matching process for each filter is independent. From the practical execution example in Section 3.2, during the programming stage we can assign a filter to several threads for execution. Figure 4 illustrates the proposed GPCDA architecture and its operating process. In Step 1, we copy the filter database from the host to the device. In Step 2, we sequentially assign the filter stored in the filter database to pre-allocated threads. Since the memory space required by the filter database is not large, we can store the filter database in a unified L1/texture cache (which is a read-only cache) to minimize memory access latency. In Step 3, each thread executes the detection algorithm independently and stores the conflict-detection results in the pre-allocated shared memory. In Step 4, each block collects the detection result reported by its associated thread and stores it in the device memory. In Step 5, all the detection results are sent back to the host through the device. We define the total execution time for conflict detection as the total time required to execute Steps 1 to 5.

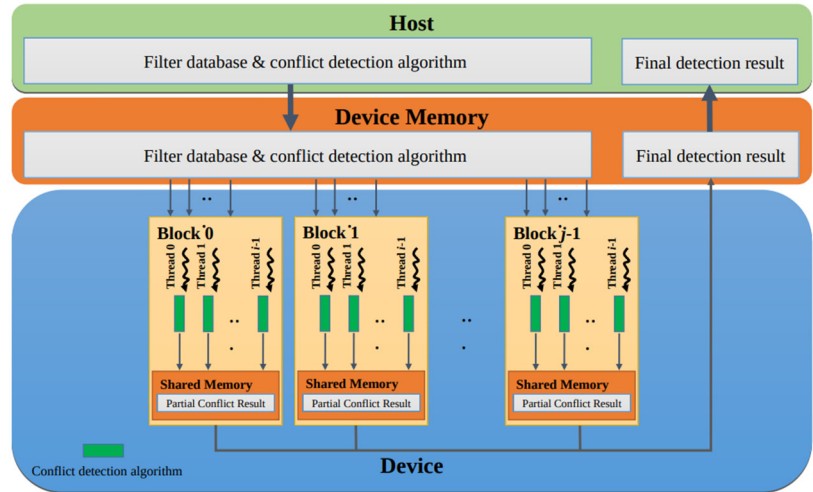

**Figure 4.** Process flow of the GPCDA.

Suppose that $i$ threads and $j$ blocks are allocated. This means that $i \times j = K$ threads are available for parallel computing. If $n$ filters should execute conflict detection, each thread must execute $\lceil n/K \rceil$ filter conflict detections. In the GPCDA, the filter dispatch order is based on the successive location in the filter database, such that threads can be assigned to execute sequentially. Assuming that $n$ filters are stored in a filter database and denoted as list $T = \{F_0, \ldots, F_{n-1}\}$, $K$ filters $F_0$–$F_{K-1}$ are initially assigned as threads 0 to $K - 1$, and the next $K$ filters, $F_K$–$F_{2K-1}$, are also assigned as threads 0 to $K - 1$. Thus, each assignment order of $K$ filters starts from thread 0 and ends at thread $K - 1$. Figure 5 illustrates how the filters in $T$ are dispatched to $K$ threads using the GPCDA. Because each filter must match other filters in $T$, each thread should execute the detection algorithm twice when $n = 2K$, and the total number of matches is $2(2K - 1)$.

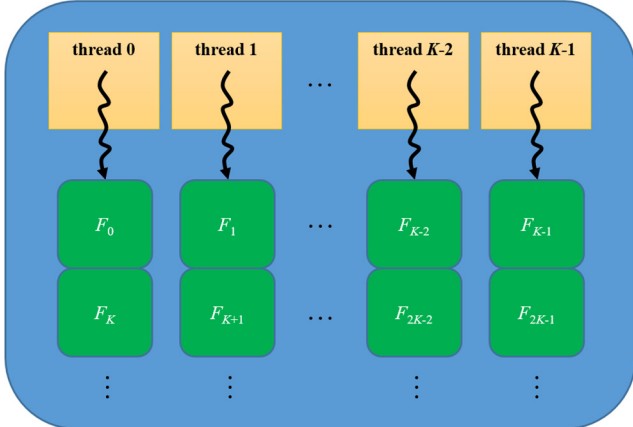

**Figure 5.** Filters dispatch of the GPCDA.

We observed that the GPCDA might obtain a duplicated detection result when the above filters perform conflict detection. Assuming that filters *f* and *g* conflict, both *f* and *g* will detect each other while executing conflict detection [33]. Therefore, we changed the matching policy. For filter $F_s$ in *T*, we only needed to compare the filter set before $F_s$, that is, filters $F_0$–$F_{s-1}$. Consequently, it could prevent the duplicated detection problem and reduce the number of comparisons required by each filter. The reduction in the number of comparisons required by each filter indicates a reduction in time for the GPCDA to execute conflict detection. This creates a new problem. If we use the term "the total number of filter comparison for each thread to execute" to represent the workload of the threads, the workload of each thread will become unbalanced (Figure 5). For example, when *n* = 2*K*, thread 0 needs to execute *K* comparisons, whereas thread *K* − 1 needs to perform a comparison 3*K* − 2 times. When the workloads between threads are unbalanced, some threads with less workload may finish the task earlier and become idle. However, threads in different blocks cannot access the same shared memory and support each other. Thus, the final detection results can only be returned to the host until the thread that "finishes the last comparison" reports its result. If the workloads among threads are balanced, thread idling and the total execution time of conflict detection can be reduced. The even distribution of the workload to threads is a key factor for improving the performance of parallelism.

*4.2. Workload Scheduling Problem*

To achieve optimized performance, we defined a workload scheduling problem regarding how to assign filters to threads evenly according to their workload.

Description of workload scheduling problem: Assuming that we perform conflict detection for *n* filters, the workloads of *n* filters are $\alpha_1$, ..., $\alpha_n$, which are assigned to *m* threads with equal computing capability. The issue is to perform workload scheduling to minimize the execution time for conflict detection.

We can convert the above problem to the well-known deterministic scheduling problem as follows: *n* independent tasks $(J_1, ..., J_n)$, which require execution times $(t_1, ..., t_n)$, respectively, are assigned to *m* processors with equal computing capability, and tasks will not be interrupted during their execution. The goal is to obtain a scheduling result so that the finish time is the shortest. During workload scheduling, the conflict detection of *n* filters is considered as an independent task. When a filter executes conflict detection, it cannot be interrupted or reassigned to another thread, and *m* threads with equal capabilities are similar to *m* processors with identical capabilities. Unfortunately, the deterministic scheduling problem has been proven to be an NP-complete problem [37,38], indicating that an optimum workload scheduling algorithm cannot be found within a limited time. Therefore, we attempted to propose a near-optimum workload scheduling algorithm. The longest processing time (LPT) algorithm [39] has been analyzed and proven to be the near-optimum algorithm closest to the optimum result.

The concept of the LPT algorithm is to arrange the task to be executed from long-to-short execution times, followed by its designed dispatch algorithm assigning tasks to the processors. The LPT algorithm guarantees that the difference in performance does not exceed $4/3 - 1/3\,m$, compared with the optimum result; that is, the lower-bound performance of the LPT algorithm is 1.33 of the optimum performance. Therefore, we propose an EPCDA based on the LPT concept. In the EPCDA, a filter list is assigned to each thread to achieve workload balancing.

### 4.3. Enhanced Parallel Conflict-Detection Algorithm

When a filter executes conflict detection, its number of comparisons represents the workload of that filter; therefore, we prearranged the order of each filter based on the number of comparisons required for that filter. In the GPCDA, each filter $F_s$ in list $T$ must be initially compared with all preceding filters in list $T$. We changed the comparison policy in the EPCDA. For each filter in the EPCDA, $F_s$ only needs to compare the filter set with the location behind $F_s$. For example, filter $F_0$ needs to be compared with $n$-1 filters behind it, and $F_1$ needs to be compared with $n$-2 filters behind it. Consequently, the number of comparisons required for each filter decreases with its sequential order in the filter list. In this way, the conflict-detection process is not duplicated, and the concept is the same as the LPT algorithm's, in which tasks are executed based on the sorted order of execution time. Furthermore, we presorted the filters in $T$ based on the SA prefix length. When the EPCDA performs a conflict detection for filters, each comparison process performs a large number of logical comparisons. The presorted order of filters ensures that the SA prefix length of each filter that is being compared does not exceed its own. This simplifies the logical comparison process and reduces the number of memory accesses. We could construct $T' = \left\{ F'_0, \ldots, F'_{n-1} \right\}$ as the list obtained after sorting $T$.

In the EPCDA, the filter dispatch order is the same as that of the method proposed in [39]. Figure 6 illustrates the EPCDA dispatch method. Initially, $K$ filters $F'_0$–$F'_{K-1}$ are dispatched to thread 0 through thread $K - 1$. Because the filters in $T'$ have been sorted based on the workload, the dispatch order of the next $K$ filters is opposite to that of the former $K$ filters to balance the workload of each thread. Thus, $F'_K$–$F'_{2K-1}$ are dispatched sequentially from thread $K - 1$ through thread 0. Whenever the EPCDA dispatches $2K$ filters, the former $K$ filters are dispatched to threads in the order from thread 0 to thread $K - 1$. In contrast, the latter $K$ filters are dispatched to threads from thread $K - 1$ to thread 0. Such a dispatch order ensures that for each $2K$ filter, the workload of each thread will be nearly balanced. Even when the number of dispatched filters is less than $2K$, it still achieves minimal difference in the workload of each thread. Algorithm 1 shows the algorithm for each thread to process filter conflict detection in the EPCDA, and Table 3 lists the notations used in Algorithm 1.

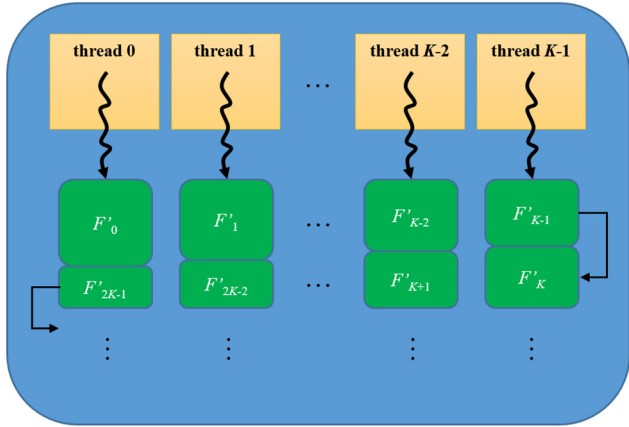

**Figure 6.** Filters dispatch of the EPCDA.

---

**Algorithm 1**: Parallel function of EPCDA

---

    **Input**:
        filter database *filter*[]
    **Output**:
        conflict results

**1**   *threadID = blockIdx.x * blockDim.x + threadIdx.x;*
**2**   *threadSize = blockDim.x * gridDim.x;*
**3**   *base = threadSize * 2;*
**4**   *start = (base − 1) − threadID;*

**5**   *i = threadID;*                    // dispatch direction →.
**6**   **do**
**7**       **for** *j ← i + 1* to *filter.size() − 1* **do**
**8**         *detection(filter[j], filter[i]);*
**9**       **end**
**10**      *i += base;*
**11**   **while** *i < filter.size()*;
**12**   *i = start;*                     // dispatch direction ←.
**13**   **do**
**14**       **for** *j ← i + 1* to *filter.size() − 1* **do**
**15**         *detection(filter[j], filter[i]);*
**16**       **end**
**17**      *i += base;*
**18**   **while** *start < filter.size()*;

---

**Table 3.** Summary of notation.

| Parameter | Notation |
|---|---|
| *blockIdx.x* | index of block |
| *theradIdx.x* | index of thread |
| *blockDim.x* | number of threads in each block |
| *gridDim.x* | number of blocks in each grid |
| *threadID* | current thread ID |

From the dispatch in the EPCDA, we observed that the numbers of comparisons required by each thread for performing conflict detection were very close. When *n* equals 2*K*, each thread must execute 2*K* − 1 comparisons. The more balanced the workload, the smaller the maximum number of total comparisons performed by the threads. In other words, the total time required to execute the detection algorithm is reduced. This finding demonstrates that the EPCDA can improve parallelism performance.

## 5. Results and Discussion

In this section, the execution performance of conflict detection based on our proposed GPCDA and EPCDA and a single CPU (Host) are evaluated. The filter databases required for the experiment were obtained from Class-Bench [40], which provided 12 parameter files obtained from three types of practical applications, including access control lists, firewalls, and IP chains. For each parameter file, we generated six filter databases with sizes 5K, 10K, 15K, 20K, 30K, and 100K. The performance evaluation indicator was defined as the average time required for each filter to execute conflict detection, counted in microseconds. (*nThread*, *nBlock*) represents the allocation of *nThread* × *nBlock* threads for parallel execution. Algorithms for simulation experiments were implemented in C++, whereas the GPCDA and the EPCDA were implemented with an additional version 7.5 CUDA toolkit for parallel programs. The test environment was set up using an Intel Core i5-4570 3.2 GHz PC with 12 GB memory. The GPU configuration was NVIDIA GeForce 970X (computation ability:

5.2) [41], which provided 13 SMs to support parallel computing. Each SM was composed of 128 SPs, 4G device memory, 98 KB shared memory, and a 48 KB L1 cache.

## 5.1. Comparing Speed Performance

In this subsection, we present a comparison of the speed performance of the Host, the GPCDA, and the EPCDA in different filter databases. The GPCDA and the EPCDA used two different total numbers of thread allocations, (256, 4) and (512, 8), for parallel computing. Figure 7 shows the required time for each filter to execute conflict detection in 12 different databases when the database size is 30K, among which databases the time for the GPCDA and the EPCDA is defined in Section 4.1. After the acceleration of parallel computing by the GPU, the GPCDA and the EPCDA are better than the Host in terms of conflict-detection speed. When the (256, 4) allocation is used, the GPCDA and the EPCDA are faster than the Host alone, by 1.3 to 4.5 and 3.9 to 9.3 times, respectively. Under the (512, 8) allocation, more threads are available to support parallel computing; thus, the speed performance significantly improves. The GPCDA and the EPCDA are 4.2 to 12.4 and 10.9 to 32.4 times faster, respectively, than the Host alone. Because the dispatch method in the EPCDA enables every thread to have a balanced workload, it improves the parallel computing performance. With the same number of threads, the EPCDA outperforms the GPCDA. The average time required for conflict detection in the EPCDA (256, 4) allocation is almost the same as in the GPCDA (512, 8) allocation, which demonstrates that workload balancing affects performance significantly.

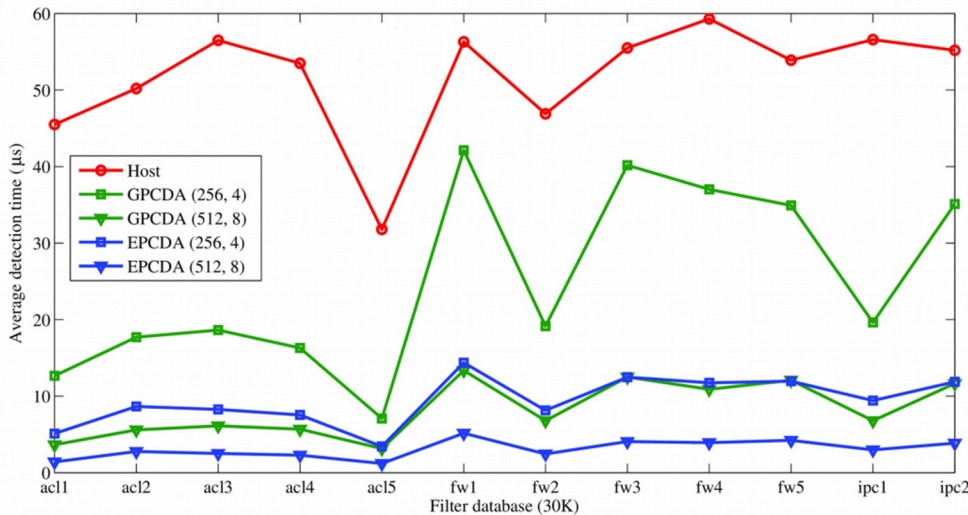

**Figure 7.** Average detection time.

Intuitively, when the number of allocated threads is doubled, performance should also be doubled. However, the experimental results (Figure 7) show that the performance did not double as expected. In the experiment, the number of allocated threads was four, but the performance improved by only 2 to 3 times. This was because when more threads executed parallel computation, the degree of resource competition increased, resulting in performance degradation.

Table 4 lists the average time required for each filter to execute conflict detection using algorithms in different databases when the filter database size is 100 K. Here, the GPCDA and the EPCDA outperformed the Host alone. In the EPCDA, the maximum time for a filter to perform conflict detection was lower than 22 μs, suggesting that the EPCDA still maintained good performance for a large database. In addition, the EPCDA can operate in applications with frequent filter updates; in such cases, fast conflict detection is required to achieve high throughputs.

**Table 4.** Performance evaluation for synthetic databases under (512, 8) allocation.

| Databases | Statistics | | Average Detection Time (µs) | | |
| --- | --- | --- | --- | --- | --- |
| | Number of Filters | Number of Conflicts | Host | GPCDA | EPCDA |
| ACL1 | 99,341 | 1,144,326 | 247.69 | 17.78 | 7.35 |
| ACL2 | 74,194 | 10,851,664 | 128.47 | 21.45 | 8.78 |
| ACL3 | 99,542 | 25,600,127 | 266.86 | 33.85 | 13.92 |
| ACL4 | 99,121 | 14,013,339 | 262.43 | 29.82 | 12.40 |
| ACL5 | 98,098 | 15 | 236.45 | 17.47 | 7.35 |
| FW1 | 87,986 | 207,072,436 | 227.10 | 70.82 | 21.79 |
| FW2 | 96,092 | 417,272,543 | 242.96 | 33.67 | 13.27 |
| FW3 | 84,472 | 227,534,381 | 207.72 | 74.92 | 20.04 |
| FW4 | 83,811 | 144,579,158 | 209.64 | 53.26 | 19.61 |
| FW5 | 83,677 | 333,717,457 | 183.99 | 65.89 | 19.65 |
| IPC1 | 99,284 | 49,864,378 | 263.68 | 35.90 | 15.42 |
| IPC2 | 100,000 | 308,440,932 | 279.19 | 52.15 | 18.91 |

*5.2. Comparison of Performance for Different Workload Dispatch Methods*

In this subsection, we evaluate the effect of different workload dispatch methods on performance. We added two workload dispatch methods for comparison with the GPCDA and the EPCDA. The first comparison used the worst workload dispatch method. Suppose that $K$ threads execute a computation. We divided $n$ filters into $K$ equal parts. The filters in each part were dispatched sequentially to the threads for execution. For example, thread 0 is responsible for performing conflict detection of filters 0 to $n/K - 1$, thread 1 is responsible for filters $n/K$ to $2n/K - 1$, and so on, until thread $K - 1$ performs conflict detection of filters $((K - 1)n)/K$ to $n - 1$. When the poorest workload dispatch method was used, we observed a significant difference in the total number of comparisons between threads 0 and $K - 1$, leading to the most significant workload imbalance. The second comparison method involved using the dispatch method of the shortest processing time (SPT) algorithm in the EPCDA (denoted EPCDA$_R$). The dispatch method in the SPT algorithm is the opposite of that in the LPT algorithm. Each filter $F_s$ was compared with filters $F_0$–$F_{s-1}$.

Here, we address the EPCDA with respect to the difference in performing conflict detection using the LPT and SPT algorithms. First, if the number of filters is a multiple of $K$, regardless of whether the LPT or the SPT algorithm is used, each thread requires the same total number of comparisons. However, if the number of filters is not a multiple of $K$, a difference exists. This trend can be explained using Figure 8. When the dispatched four threads performed conflict detection of filters $F_0$–$F_8$, the filters dispatched in the LPT algorithm experienced a significantly more balanced workload than those in the SPT algorithm. The total number of comparisons for both algorithms was 9. For example, $F_0$ and $F_7$ required eight and one comparisons, respectively, in thread 0 ($F_8$ did not need an additional comparison), and $F_2$ and $F_5$ required six and three comparisons, respectively, in thread 2. However, when filters were dispatched through the SPT algorithm, the workload in thread 0, which reached 15 (needing 0, 7, and 8 comparisons) was higher than that of the other threads. Workload imbalance increased the total execution time for conflict detection.

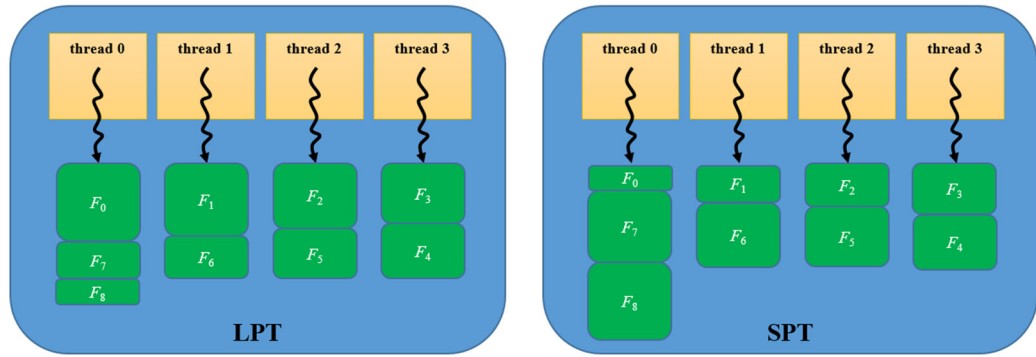

**Figure 8.** Filter dispatch for different dispatch methods (LPT and SPT).

Second, for the number of memory access times, after the dispatch of the LPT algorithm, while processing the first conflict detection in each thread, each thread wrote the compared filters in the cache, owing to a cache miss. Assuming that the cache size is sufficiently large to store all compared filters, the next conflict-detection process could be immediately performed because the filters to be compared were already written to the cache. For example, during $F_0$ conflict-detection processing, thread 0 needed to load filters $F_1$–$F_8$ into its local cache because of cache miss; therefore, there were no additional memory access times during $F_7$ and $F_8$ conflict-detection processing. In contrast, most comparisons in the SPT algorithm still indicated cache misses after the first conflict detection was processed. Consequently, it needed to wait until the filter to be compared was written to the cache, resulting in a significant amount of memory access and an increase in the total required time for conflict detection. Under a highly unbalanced workload, the parallel performance was significantly reduced (Figure 9). The average execution time for conflict detection in each filter exceeded that in the GPCDA. Although the performance was better than that of the GPCDA when a filter used the EPCDA$_R$ to execute conflict detection, the EPCDA was the best when the LPT algorithm was used to dispatch tasks.

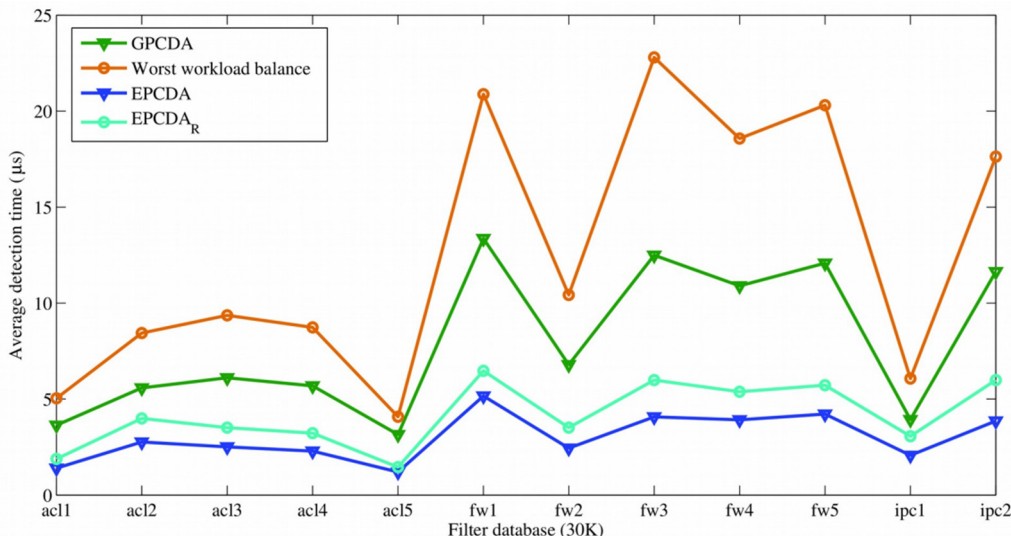

**Figure 9.** Comparison of workloads.

### 5.3. Analysis of Parallelism Efficiency

In this subsection, we discuss the efficiency of parallelism for a fixed total number of threads using different (*nThread*, *nBlock*) allocations for parallel computing. Here, we compare the efficiency of the (256, 8) and (512, 4) allocations. Tables 5 and 6 list the average required times to execute conflict detection by a filter using the Host, the GPCDA, and the EPCDA, with databases of the lowest and highest numbers of detected conflicts for

the three filter database types. The average required times for a filter to execute conflict detection in the GPCDA and the EPCDA were several times lower than that in the Host alone, regardless of the condition of the smallest or largest number of conflicts.

**Table 5.** Average detection time (µs) for databases with least conflicts in each category.

| Database | Algorithm | Filter Size | | | | |
|---|---|---|---|---|---|---|
| | | **5K** | **10K** | **15K** | **20K** | **30K** |
| ACL5 | Host | 2.94 | 10.91 | 13.69 | 21.08 | 31.91 |
| | GPCDA (256, 8) | 0.72 | 1.61 | 2.06 | 2.86 | 3.97 |
| | GPCDA (512, 4) | 0.90 | 1.92 | 2.52 | 3.40 | 4.73 |
| | EPCDA (256, 8) | 0.34 | 0.72 | 0.88 | 1.37 | 1.72 |
| | EPCDA (512, 4) | 0.35 | 0.82 | 1.00 | 1.56 | 1.97 |
| FW4 | Host | 10.45 | 20.85 | 30.73 | 40.11 | 59.46 |
| | GPCDA (256, 8) | 4.10 | 7.55 | 9.20 | 11.85 | 16.63 |
| | GPCDA (512, 4) | 4.74 | 8.23 | 10.69 | 13.11 | 20.00 |
| | EPCDA (256, 8) | 2.29 | 2.51 | 3.64 | 4.35 | 6.00 |
| | EPCDA (512, 4) | 2.71 | 2.93 | 4.26 | 5.14 | 7.08 |
| IPC1 | Host | 8.82 | 19.06 | 28.18 | 37.57 | 56.53 |
| | GPCDA (256, 8) | 1.82 | 3.77 | 5.35 | 6.42 | 10.03 |
| | GPCDA (512, 4) | 2.23 | 4.48 | 6.38 | 7.64 | 12.17 |
| | EPCDA (256, 8) | 0.96 | 2.12 | 2.58 | 3.21 | 4.72 |
| | EPCDA (512, 4) | 1.12 | 2.46 | 3.01 | 3.70 | 5.45 |

**Table 6.** Average detection time (µs) for databases with most conflicts in each category.

| Database | Algorithm | Filter Size | | | | |
|---|---|---|---|---|---|---|
| | | **5K** | **10K** | **15K** | **20K** | **30K** |
| ACL2 | Host | 8.69 | 18.19 | 26.16 | 34.83 | 50.10 |
| | GPCDA (256, 8) | 1.88 | 3.34 | 4.51 | 6.60 | 9.20 |
| | GPCDA (512, 4) | 2.42 | 4.22 | 5.65 | 8.16 | 11.26 |
| | EPCDA (256, 8) | 0.87 | 1.87 | 2.54 | 3.29 | 4.36 |
| | EPCDA (512, 4) | 1.00 | 2.18 | 3.02 | 3.86 | 5.16 |
| FW2 | Host | 7.70 | 16.80 | 25.08 | 33.41 | 49.57 |
| | GPCDA (256, 8) | 1.80 | 3.69 | 5.35 | 6.46 | 9.92 |
| | GPCDA (512, 4) | 2.38 | 4.79 | 6.68 | 7.98 | 12.62 |
| | EPCDA (256, 8) | 0.90 | 1.78 | 2.25 | 2.81 | 4.32 |
| | EPCDA (512, 4) | 1.04 | 2.04 | 2.55 | 3.21 | 4.89 |
| IPC2 | Host | 8.64 | 18.57 | 27.73 | 36.94 | 55.20 |
| | GPCDA (256, 8) | 3.28 | 5.12 | 8.87 | 10.31 | 17.48 |
| | GPCDA (512, 4) | 3.40 | 5.54 | 9.37 | 11.25 | 18.51 |
| | EPCDA (256, 8) | 1.72 | 2.97 | 3.57 | 4.43 | 6.41 |
| | EPCDA (512, 4) | 1.95 | 3.35 | 3.97 | 4.98 | 7.42 |

Regardless of whether the GPCDA or the EPCDA was used, the time required for each filter to execute conflict detection under the (256, 8) allocation was always shorter than that under the (512, 4) allocation. This trend indicates that for the same total number of threads, the more threads in a block, the worse the parallel performance. Through analysis with NVIDIA Visual Profiler [42], two major factors that affect parallelism efficiency are "stalled for memory dependency" and "stalled for synchronization". "Memory dependency" is mainly attributed to the additional stall caused by the data dependency of two consecutive instructions, whereas "stalled for synchronization" is mainly attributed to the _syncthreads( ) instruction in the CUDA. Recall that threads in a block use a warp as a unit for hardware to execute parallel computing, and threads in the same block access the data in the same shared memory. When threads in the same warp are executed, the CUDA invokes the _syncthreads( ) instruction to ensure data consistency while the threads

are performing computations. If tasks in some threads of the same warp are completed earlier, the threads must wait for other threads to complete their tasks, and that batch of threads proceeds to the next dispatched task. Therefore, when a block is dispatched with a higher number of threads, it is easier to cause "stalled for synchronization" and affect parallel efficiency

### 5.4. Limitations of Parallel Computation

Based on the results discussed thus far, it can be observed that when the number of allocated threads and blocks increases, the performance improves, so that the GPCDA and the EPCDA can handle a large filter database. In this subsection, we discuss the limitations of parallel computation in a large-filter database in the GPCDA and the EPCDA. When the number of filters was large, even when the number of allocated threads for parallel computing increases several times, the performance did not increase accordingly (Figure 10). We have discussed the reason for this above. We observed that for parallel computing of threads under allocation 13,312 (1024, 13), the GPCDA and the EPCDA exhibited the best performance. However, the performance for threads under allocation 16,384 (1024, 16) decreased. GTX 970 supports 13 SM at most for parallel computing, and each block allows only up to 1024 threads. If more blocks are allocated, it causes some SMs to execute at least two computation blocks, and the execution time increases, due to hardware scheduling. In the 8192 (1024, 8) allocation, when threads executed the EPCDA, the difference was minimal compared to configuration 13,312 (1024, 13). Thus, when the workload approached a balance, we could use less hardware to achieve the same performance. This confirmed that the EPCDA can execute conflict detection using a GPU with a few cores to reduce hardware costs.

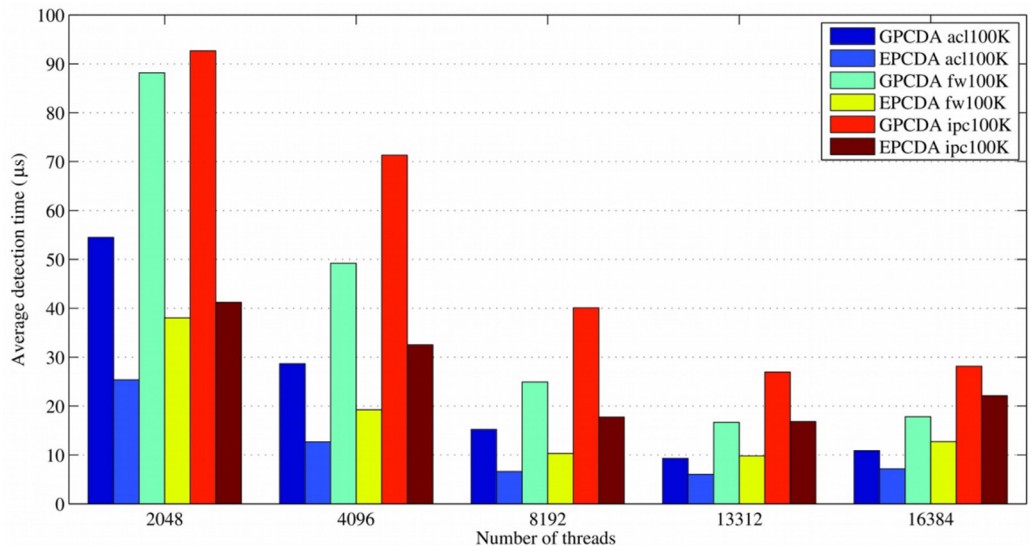

**Figure 10.** Limitation of parallel computing.

### 6. Conclusions

In this study we applied GPUs to conflict-detection algorithms, and developed the GPCDA and the EPCDA, which can accelerate conflict detection through parallel computing in GPUs. Based on the simulation experiment, we found that the GPCDA and the EPCDA performed conflict detection up to 33.7 times faster than the CPU, regardless of the type and size of the filter databases; this was particularly evident when the number of allocated threads for parallel computing became large. We also observed that for the same total number of threads, different numbers of allocated blocks and threads significantly influenced the parallel efficiency. We analyzed the factors affecting the parallel efficiency, which may guide GPU utilization in other applications.

If the workload between threads was unbalanced, different threads required different computation times. This may have caused some threads to be idle because of a smaller workload, whereas threads with a larger workload spent more time, increasing the total parallel computing. Therefore, the workload balance was an essential factor that influenced parallel performance. Unfortunately, the literature reports that dispatching tasks with different execution times to multicores with the same computing capability and achieving optimized workload balance is an NP-complete problem. Thus, we developed a near-optimum workload balance mechanism in the EPCDA, such that the workload among threads could approach a balance and maximize the performance of parallel computing. Based on the simulation experiment, we observed different performances for different workload dispatch methods. Using the EPCDA, even though we used fewer threads for parallel computing, it achieved the same performance as the GPCDA, which used more threads for parallel computing; thus, the hardware cost was reduced.

Finally, when the number of filters was large, the EPCDA still maintained good performance, with a filter database size of 100K. In the EPCDA, each filter spent 22 µs, at most, to process conflict detection; this was 9.4 to 33.7 times faster than using only a CPU. Therefore, the EPCDA is suitable for applications with frequent filter-database updates. The limitations and future research directions of this study are as follows. First, the detection speed can be increased by improving the GPU memory access efficiency, such as by minimizing non-coalesced memory accesses and bank conflicts. Second, in this study, we focused on analyzing the critical procedure in 5D conflict detection, and balancing the workload of GPU threads. The proposed algorithms were not designed for a specific GPU architecture/model. For a specific GPU architecture/model, our proposed algorithms can be modified to achieve better performance by taking advantage of hardware features. Third, this study assumed that conflict detection is executed on a single-GPU platform. For a heterogeneous multi-GPU platform, the workload balancing problem becomes much more complicated and the communication cost between GPUs should be taken into account when designing a conflict-detection algorithm.

**Author Contributions:** Conceptualization, C.-L.L. and Y.-C.C.; data curation, G.-Y.L.; formal analysis, G.-Y.L.; investigation, G.-Y.L.; methodology, G.-Y.L.; project administration, C.-L.L. and Y.-C.C.; resources, Y.-C.C.; software, G.-Y.L.; supervision, C.-L.L. and Y.-C.C.; validation, G.-Y.L.; writing—original draft, G.-Y.L.; writing—review and editing, C.-L.L. and Y.-C.C. All authors have read and agreed to the published version of the manuscript.

**Funding:** This research was funded by the Ministry of Science and Technology of Taiwan (MOST 104-2221-E-182-005 and 110-2221-E-182-010) and Chang Gung Memorial Hospital (BMRP 942).

**Data Availability Statement:** Not applicable.

**Conflicts of Interest:** The authors declare no conflict of interest.

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
