# Peer review of "An Efficient Parallel Algorithm for Detecting Packet Filter Conflicts"

_algorithms, doi:10.3390/a15070237_

Round 1
Reviewer 1 Report
You are advised to refer to few more journals papers published recently i.e. from 2022, prefer high indexed journals like SIAM, Transaction etc. Accordingly, improve the review of literature.
Application of the work done in the paper need more extensive discussion, along with challenges during the deployment.
Reviewer 2 Report
This is avery interesting article that worth's publication. There are some points that would should be taken into account
1) Include ref
M. Abbasi & M. Rafiee, A calibrated asymptotic framework for analyzing packet classification algorithms on GPUs, The Journal of Supercomputing volume 75, pages 6574–6611 (2019)
in the introduction for completeness
2) Is there any dependence on the model of GPU employed on the performance of the algorithms? Could the authors discuss o this?
Reviewer 3 Report
This paper two parallel implementations on GPU architectures that can solve five-dimensional (5D) filter conflict problems. These implementations are parallelized using CUDA and achieves superior performance for several filter datasets. This computational work also is considered great importance for researchers in network services.
The paper is well written and easy to follow. The authors have conducted comprehensive experiments to evaluate their proposed approach. However, I have the following suggestions.
1) The authors report differences of the proposed approaches compared to other works.
2) The conclusions do not speak about the limitations of the research and do not indicate the perspective of future research plans.
Round 2
Reviewer 3 Report
The manuscript is revised taking into account the recommendations made in the review.